# TRANSFER LEARNING FOR BAYESIAN HPO WITH END-TO-END META-FEATURES

## ABSTRACT

Hyperparameter optimization (HPO) is a crucial component of deploying machine learning models, however, it remains an open problem due to the resource-constrained number of possible hyperparameter evaluations. As a result, prior work focuses on exploring the direction of transfer learning for tackling the sample inefficiency of HPO. In contrast to existing approaches, we propose a novel **D**eep **K**ernel Gaussian Process surrogate with **L**andmark **M**eta-features (DKLM) that can be jointly meta-trained on a set of source tasks and then transferred efficiently on a new (unseen) target task. We design DKLM to capture the similarity between hyperparameter configurations with an end-to-end meta-feature network that embeds the set of evaluated configurations and their respective performance. As a result, our novel DKLM can learn contextualized dataset-specific similarity representations for hyperparameter configurations. We experimentally validate the performance of DKLM in a wide range of HPO meta-datasets from OpenML and demonstrate the empirical superiority of our method against a series of state-of-the-art baselines.

## 1 INTRODUCTION

Hyperparameter optimization (HPO) is an essential open problem in machine learning (ML) due to the black-box nature of methods' empirical performances as a function of their hyperparameters. The major challenge lies in the computational infeasibility of training and evaluating a large sample of hyperparameters in order to identify the best generalization performances. As a result, transfer learning lends itself as a promising direction for improving the sample efficiency of HPO methods (Wistuba et al., 2016; Perrone et al., 2018; Wistuba & Grabocka, 2021).

Prior approaches for transfer learning in HPO rely on exploring existing evaluations on a pool of datasets where the model under investigation is evaluated. The similarity between datasets is often captured via features describing their characteristics (a.k.a. meta-features), such as descriptive statistics of dataset features (Michie et al., 1994; Wistuba et al., 2016), or landmark measures in the form of the accuracies gathered from a set of basic classifiers (nearest neighbors, decision trees, SVMs, etc.) on the datasets (Pfahringer et al., 2000; Feurer et al., 2014). A recent trend highlights the potential of learning parametric meta-feature extractors for tabular datasets (Jomaa et al., 2021a), which are further meta-trained (Finn et al., 2017) to improve the HPO transferability to new datasets (Jomaa et al., 2021b).

Unfortunately, typical transfer learning from a set of unrelated source tasks suffers from the negative transfer phenomenon (Wang et al., 2019), which implies a poor generalization performance on target tasks that are dissimilar to the source tasks, according to a predefined dissimilarity measure, e.g. similarity of response curves. This can happen, for example, when a model is learned jointly across tasks without task-specific attributes. To resolve the negative transfer of HPO performance predictors (a.k.a. surrogates) we introduce a novel direction that conditions Gaussian Process (GP) surrogates in Bayesian Optimization on the meta-features of datasets. In that manner, we can transfer knowledge only from similar datasets, and hence conditioned on the similarity of meta-features. However, in contrast to ad-hoc dataset meta-features that are hand-crafted by domain experts, we propose a novel architecture for deep GP kernels (Wilson et al., 2016a) that are enriched with novel end-to-end neural network components that generate meta-features only from the tuples of past hyperparameter configurations and their evaluated performances. Our meta-feature network

is a set-based neural network that is invariant to the permutation/sequence of past hyperparameter evaluations.

We jointly train **D**eep **K**ernel Gaussian Process surrogate with **L**andmark **M**eta-features (DKLM) through HPO meta-learning (Wistuba & Grabocka, 2021). To validate the empirical performance of our method we present extensive results on a large-scale benchmark that involves 16 different search spaces and 101 datasets from OpenML for a total of 3.4 million hyperparameter evaluations (Pineda-Arango et al., 2021). Detailed experiments against a series of traditional HPO methods, as well as recent transfer HPO baselines, demonstrate the superiority of meta-learning the initialization of DKLM. Overall, we make the following contributions:

- Introduce the first paper that tackles the negative transfer phenomenon in Bayesian HPO, by conditioning GP surrogates on meta-features, i.e. on dataset characteristics;

- Propose an end-to-end deep GP which implicitly learns networks that generate meta-features, with no ad-hoc inductive bias from experts on manually designing meta-features;

- Demonstrate the empirical superiority of our method on a very-large-scale experimental protocol (3.4 million hyperparameter evaluations), against a large number of baselines.

## 2 RELATED WORK

Hyperparameter optimization (HPO) has been extensively studied over the past decade for improving the performance of machine learning models beyond simple search techniques (Larochelle et al., 2007; Bergstra & Bengio, 2012). Non-transfer learning solutions often define a probabilistic surrogate that estimates the true hyperparameter response surface using Gaussian Processes (Rasmussen, 2003), Bayesian Neural Networks (Snoek et al., 2015; Springenberg et al., 2016), or tree-based models (Hutter et al., 2011; Bergstra et al., 2011). Hyperparameters are then selected via an acquisition function (Wilson et al., 2017) and the process is reiterated with the new set of observations until a specified budget is exhausted, e.g. runtime or number of trials.

HPO is further expedited when defined within the context of transfer learning, i.e. by leveraging related tasks (or datasets) to improve the generalization over unseen tasks. Transfer learning for HPO has been observed by modeling tasks jointly (Swersky et al., 2013; Yogatama & Mann, 2014; Perrone et al., 2018; Salinas et al., 2020), or through some weighted-combination of the surrogates (Schilling et al., 2016; Wistuba et al., 2016; Feurer et al., 2018). Other directions include pruning the hyperparameter search space (Wistuba et al., 2015a; Perrone & Shen, 2019), or learning to initialize the surrogate by identifying good initial hyperparameters (Wistuba et al., 2015b). Apart from learning a transferable surrogate, recently, transferable acquisition functions (Wistuba et al., 2018; Volpp et al., 2020) have also been proposed to replace engineered acquisition functions. The success of meta-learning for domain adaptation has also been investigated for HPO. Wistuba & Grabocka (2021) explore few-shot Bayesian Optimization by learning a deep kernel Gaussian Process surrogate across a set of tasks to quickly adapt to new a target task. Similarly, Jomaa et al. (2021b) learn a shared neural network surrogate jointly coupled with a meta-feature extractor defined over the dataset itself.

Meta-features (Vanschoren, 2018), or dataset characteristics, have also been widely adopted in HPO algorithms for warm-start initialization (Feurer et al., 2015; Wistuba et al., 2016) or as additional attributes to better marginalize the surrogate on individual tasks (Bardenet et al., 2013). Nevertheless, extracting meta-features requires direct access to the datasets, which might be difficult in real settings where only the meta-dataset is available. In this paper, we propose to extract landmark meta-features from existing evaluations (Leite et al., 2012; Sun & Pfahringer, 2013) in an end-to-end fashion using a deep Gaussian kernel approach.

## 3 PRELIMINARIES

### 3.1 HYPERPARAMETER OPTIMIZATION

We denote by $\mathcal{D} = \{(x_i, y_i)\}_{i=1}^n$ a task of interest, such that $x_i \in \mathcal{X} \subseteq \mathbb{R}^n$ represents a hyperparameter configuration in the domain of a (bounded) hyperparameter search space for some model

under investigation. Furthermore, let $y_i = f(x_i) + \epsilon$ be the response of an unknown black-box function $f := \mathcal{X} \to \mathbb{R}^+$, with $\epsilon$ as an additive i.i.d Gaussian noise with some homoscedastic variance $\sigma^2$. Typically, $y$ represents a metric of interest that should be optimized to obtain better model generalization, e.g. validation loss. The objective of hyperparameter optimization is then to find the optimal hyperparameter such that $x_* = \arg\min_{x \in \mathcal{X}} f(x)$ given a fixed budget $T$ of trials. HPO is commonly treated as sequential decision-making process, where a surrogate model $\hat{y} : \mathcal{X} \to \mathbb{R}$ is iteratively fit to the history $\mathcal{H}_t := \{(x_i, y_i)\}_{i=1}^t$ of evaluated hyperparameters and a policy (or acquisition function) $\mathcal{A} : (\mathcal{X} \times \mathbb{R})^* \to \mathcal{X}$ is used to select the next candidate which minimizes the expected hyperparameter response. Among the existing acquisition functions, *expected improvement* is widely adopted (Mockus, 1974).

### 3.2 DEEP KERNEL GAUSSIAN PROCESSES

Given a training task $\mathcal{D} = \{(x_i, y_i)\}_{i=1}^n$, the response can be modeled using a Gaussian process (GP), i.e. as a multivariate Gaussian distribution, such that $y \sim \mathcal{N}(m(X), k(X, X))$. A GP is a non-parametric approach that defines a prior over functions directly, and is defined by its mean function, $m$, and kernel function $k$. Given some observed data points, it is possible to compute the posterior over these functions to approximate unobserved data points as

$$\begin{bmatrix} y \\ f_* \end{bmatrix} \sim \mathcal{N}\left(m(X), \begin{pmatrix} K_n & K_* \\ K_*^T & K_{**} \end{pmatrix}\right) \tag{1}$$

with $K_n = k(X, X \mid \theta) + \sigma_n^2 \mathbb{I}$, $K_* = k(X, X_* \mid \theta)$, and $K_{**} = k(X_*, X_* \mid \theta)$. The mean and covariance of the posterior predictive distribution is then estimated as

$$\mathbb{E}[f_* \mid X, y, X_*] = K_*^T K_n^{-1} y, \ \text{cov}[f_* \mid X, X_*] = K_{**} - K_*^T K_n^{-1} K_* \tag{2}$$

The standard approach of fitting GPs is to optimize the weights of the kernel function, e.g. squared exponential kernel, $\theta$. Nevertheless, these engineered kernels are often employed under false assumptions (Cowen-Rivers et al., 2020), which leads to sub-optimal performances.

Recently deep kernel learning (Wilson et al., 2016b) has emerged as a powerful extension that leverages the representative capacity of non-linear function approximation, e.g. neural networks, and facilitates learning the kernel directly. Specifically, we denote by $\phi : \mathcal{X} \to \mathbb{R}^N$ a mapping from the domain to a latent space which serves as an input to the kernel, such that:

$$K_{\text{deep}}(x, x' \mid \theta, w) = K(\phi(x, w), \phi(x', w) \mid \theta) \tag{3}$$

where $w$ represents the parameters of $\phi$. The weights $\theta$ and $w$ are then jointly optimized for maximizing the marginal likelihood (Wistuba & Grabocka, 2021).

## 4 DEEP KERNEL GAUSSIAN PROCESS WITH LANDMARK META-FEATURES

Inspired by landmark meta-features (Pfahringer et al., 2000), which are typically estimated by measuring the response of given datasets to machine learning algorithms, we propose a novel deep kernel GP that is conditioned on task-specific landmark meta-features. However, instead of computing meta-features through ad-hoc approaches, we introduce a novel parametric meta-feature extractor network that is integrated into the kernel function of a GP and subsequently meta-learned over a set of source tasks together with the parameters of the GP kernel. In that manner, we learn meta-features that describe a set of tasks in terms of minimizing the estimation of the tasks' response functions. By adding the task-specific information of the meta-features, the GP surrogate can infer a more accurate response surface on a new task based on similar source tasks that share similar meta-features. Therefore, our method is the first to tackle the negative transfer phenomenon for Bayesian HPO.

### 4.1 LANDMARK META-FEATURE NETWORKS

We propose a simple idea to learn landmark meta-features by learning a deep representation of the evaluated **set** of hyperparameter-response pairs as part of a deep kernel Gaussian process. With the success of set-based algorithms (Lee et al., 2020; Zaheer et al., 2017; Lee et al., 2019) for function

approximation, we propose to use a *Deepset* (Zaheer et al., 2017) formulation that provides a fixed-size vector representation from the dynamic set of observations. Although other methods have been developed for set-based function estimation, we focus here on deepsets because they have already been shown to perform well for learning meta-feature in task-agnostic settings (Jomaa et al., 2021a) as well as for hyperparameter optimization (Jomaa et al., 2021b).

Suppose that we are given a collection of data points $\{(x_i, y_i)\}_{i=1}^n$ where $x \in \mathcal{X}$ is an observed hyperparameter and $y \in \mathbb{R}$ its corresponding response. We denote by $\mathcal{H}^{t-1} := \{(x_i, y_i)\}_{i=1}^{t-1}$ an associated set of data points that have been observed prior to $x_t$. Furthermore, we formulate the proposed meta-feature network as:

$$\phi(x, \mathcal{H}^{t-1}, w) = \phi_1 \left( [x, \phi_2(\mathcal{H}^{t-1}; w_{\phi_2})]; \ w_{\phi_1} \right), \tag{4}$$

$$\text{s.t. } \phi_2(\mathcal{H}^{t-1}; w_{\phi_2}) = g \left( \frac{1}{t-1} \sum_{i=1}^{t-1} f\left([x_i, y_i]; \ w_f\right); \ w_g \right) \tag{5}$$

where $[\ ]$ symbolizes standard concatenation, $\phi_2 : (\mathcal{X} \times \mathbb{R})^* \to \mathbb{R}^N$ and $\phi_1 : \mathcal{X} \times \mathbb{R}^N \to \mathbb{R}^M$ are parametric neural networks with respective weights $w$, and where $(\mathcal{X} \times \mathbb{R})^*$ represents the set of evaluated hyperparameter and their responses. With this formulation, we ensure that the relationship of the covariates and the responses in $\mathcal{H}^{t-1}$ is properly encoded, and thus $\phi$ is conditioned on these latent representations. It is also important to note that $\phi_2$ is permutation invariant, i.e. $\phi_2(\mathcal{H}^{t-1}) = \phi_2(\pi(\mathcal{H}^{t-1}))$, with $\pi := (\mathcal{X} \times \mathbb{R})^* \to (\mathcal{X} \times \mathbb{R})^*$ as a random permutation function. This is critical, as the ordering of the data points in $\mathcal{H}^{t-1}$ should not affect the landmark meta-features. Additionally, given $\phi_2(\mathcal{H}^{t-1})$, this information about the marginal distribution of the meta-features can be encoded with the specific attribute $x$ which in turn allows the deep kernel GP to be marginalized over individual tasks given the context, and thus transfer (joint) learning becomes easier with minimal overhead.

## 4.2 META-LEARNING OUR DEEP GPS

The parameters $\theta$ and $w$ are optimized jointly by maximizing the following log marginal likelihood:

$$\underset{\theta, w}{\arg\max} \ \log p\left(\mathbf{y} \mid x, \mathcal{H}; \theta, w\right) = \underset{\theta, w}{\arg\max} \ \mathbb{E}_{d \sim \mathcal{U}(1, \dots, D)} \ \log p\left(\mathbf{y}_d \mid x_d, \mathcal{H}_d; \theta, w\right) \tag{6}$$

$$\propto \underset{\theta, w}{\arg\min} \ \mathbb{E}_{d \sim \mathcal{U}(1, \dots, D)} \ \mathbf{y}_d^T K_d^{-1} \mathbf{y}_d + \log |K_d| \tag{7}$$

$$\text{s.t. } K_{d,t,t'} := K\left(\phi\left(x_{d,t}, \mathcal{H}_d^{t-1}; w\right), \phi\left(x_{d,t'}, \mathcal{H}_d^{t'-1}; w\right); \theta\right)$$

By using established practices (Wistuba & Grabocka, 2021; Patacchiola et al., 2020), we can optimize Equation 7 in terms of $w, \theta$ via stochastic gradient descent (SGD), that is proven to maintain convergence guarantees (Chen et al., 2020). We direct the interested reader to the prior work for more details on optimizing the parameters of deep GPs (Wilson et al., 2016b).

Given the diverse number of tasks, which vary in the number of available data points, we propose to jointly learn the shared surrogate via first-order meta-learning (Nichol et al., 2018). Meta-learning has found resounding success in the research community as an initialization scheme (Finn et al., 2017; Wistuba & Grabocka, 2021; Jomaa et al., 2021b), which allows for fast adaption to new domains. Consequently, the meta-trained model resides on a joint minimum across all the source tasks, such that given limited information about the new (unseen) target task, it can converge faster to the new task's local optima. In this direction, we show the pseudocode of our meta-learning optimization in Algorithm 1.

## 5 MOTIVATION

**Meta-features help to model the posterior uncertainty.** To motivate our approach, we present an ablation of the effect of our deep GP kernel with meta-features, compared to the same deep GP kernel without meta-features (i.e. Ours vs. FSBO (Wistuba & Grabocka, 2021)). We created a synthetic meta-dataset of $K = 50$ tasks in the form of randomly sampled sinusoidal functions

---

**Algorithm 1:** Meta-learning DKLM via REPTILE (Nichol et al., 2018)

1: **Require:** training dataset $\mathcal{E}$; kernel parameters $\theta$, network parameters $w$; learning rate $\eta$; inner update steps $v$; meta-batch size $n$, batch size $b$.
2: **while** not converged **do**
3:     $t \sim \mathcal{U}\left([T_{\min}, T_{\max}]\right)$
4:     $D_1, \ldots, D_n \sim \mathcal{U}\left([1, \ldots, D]\right)$
5:     **for** $i = 1$ to $n$ **do**
6:         Sample $t - 1$ data points to form $\mathcal{H}^{t-1} \sim \mathcal{D}_i$
7:         Sample batch $\mathcal{B} := \{(x_i, y_i)\}_{i=1}^b \sim D_i$
8:         $\theta_i \leftarrow \theta \; ; w_i \leftarrow w$
9:         **for** $j = 1$ to $v$ **do**
10:             Define as $\mathcal{L}$ the objective function of Equation 7
11:             $\theta_i \leftarrow \theta_i + \eta \nabla_\theta \mathcal{L}$
12:             $w_i \leftarrow w_i + \eta \nabla_w \mathcal{L}$
13:     Update $\theta \leftarrow \theta + \eta \frac{1}{n} \sum_{i=1}^n (\theta_i - \theta)$
14:     Update $w \leftarrow w + \eta \frac{1}{n} \sum_{i=1}^n (w_i - w)$

---

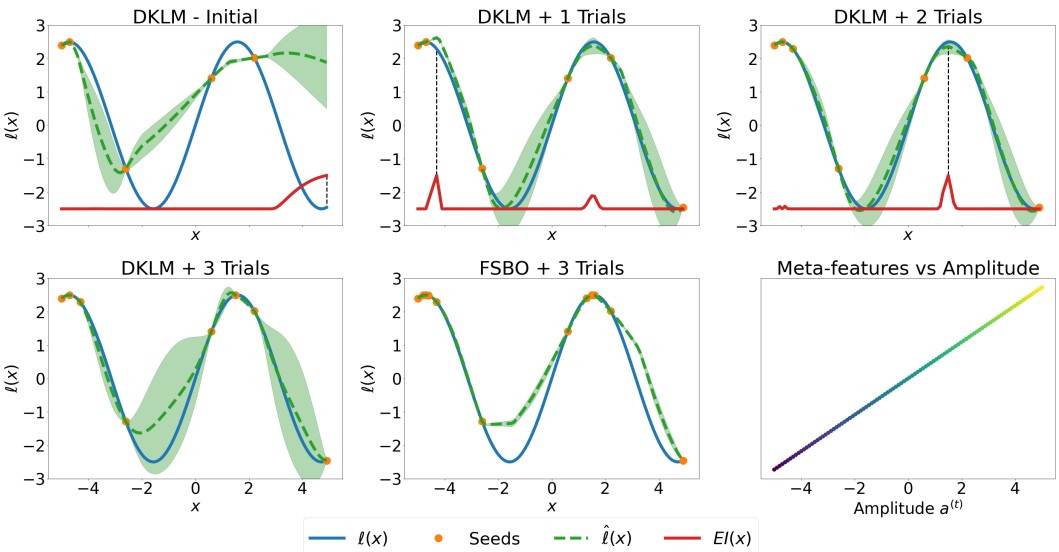

Figure 1: (top) Sequential model-based optimization of an unseen sine wave using our approach. (bottom left and middle) Our fitted surrogate after 3 trials, compared to FSBO after 3 trials given the same initial seeds. (bottom right) Correlation between amplitude and landmark meta-features.

$f^k(x) = a^{(k)} \sin(x + b^{(K)})$, $k \in \{1, \ldots, K\}$ by drawing each $a^{(k)} \sim \mathcal{U}(0.1, 5)$ and $b^{(k)} \sim \mathcal{U}(0, 2\pi)$. Furthermore, we meta-learn our deep GP on these source functions and then transfer the surrogate as an initialization for a new sinusoidal curve (with new parameters $a, b$) as shown in Figure 1 (top). We show the comparison of our surrogate with meta-features after 3 trials (for a total of 8 data points, including 5 initial configurations) to an FSBO deep GP that has been meta-trained identically. We notice that our surrogate (bottom row, leftmost plot) computes a better posterior variance compared to FSBO (bottom row, middle plot). The effect of the superior modeling of the uncertainty leads to better exploration in a Bayesian Optimization setup, and consequently to better empirical accuracies of the discovered hyperparameters (as will be shown in Section 6).

**Meta-features capture task characteristics.** We postulated that our proposed meta-features can capture task characteristics. To illustrate the argument, we create another simpler collection of $K = 50$ sinusoidal functions $f^k(x) = a^{(k)} \sin(x)$, $k \in \{1, \ldots, K\}$ by drawing each $a^{(k)} \sim \mathcal{U}(0.1, 5)$. As these sine waves change only in terms of the amplitude parameter $a$, then if we meta-train our meta-feature network with a 1-dimensional (1D) output layer from these source functions, it must strongly learn to correlate the 1D meta-feature with the sinusoidal amplitude $a$. As can

be seen in the rightmost plot of the bottom row in Figure 1, this is exactly the case. In this plot, the y-axis shows 1D meta-feature values computed from only 5 random pairs of configurations and responses $(x, f(x))$ from one random task, and the x-axis shows the amplitude of that respective task. Although our meta-feature networks have no design bias in terms of modeling sinusoidal functions, they are perfectly able to extract a latent representation of the amplitude, based on the end-to-end meta-learning of the deep GP for approximating random observations on the sine waves.

## 6 EXPERIMENTS

Our experimental protocol is designed to primarily address one simple research question: **Do deep GPs with our meta-feature networks outperform state-of-the-art HPO algorithms in the transfer and non-transfer learning settings?**

### 6.1 META-DATASET AND BASELINES

We evaluate our approach on HPO-B-v3, a new hyperparameter optimization benchmark designed for comparing black-box HPO methods (Pineda-Arango et al., 2021). The benchmark contains a collection of 935 black-box tasks for 16 hyperparameter search spaces (algorithms) evaluated on 101 datasets and divided into predefined training, validation, test splits. Following the same experimental protocol specified at the HPO-B benchmark, we compare our approach to the following large set of 10 HPO baselines:

1. **Random Search** (Bergstra & Bengio, 2012);

2. **GP** (Rasmussen, 2003) is a hyperparameter tuning strategy that relies on a Gaussian Process as a surrogate model with squared exponential kernels (Matern 5/2 kernel) with automatic relevance determination;

3. **DNGO** (Snoek et al., 2015) utilizes a neural network to extract adaptive basis function of hyperparameters, which in turn are fed to a Bayesian linear regression model to generate a posterior distribution;

4. **BOHAMIANN** (Springenberg et al., 2016) is based on Bayesian neural networks that are trained via a stochastic gradient Hamiltonian Monte Carlo;

5. **DGP** (Patacchiola et al., 2020) fits a deep kernel Gaussian process as a surrogate;

6. **TST-R** (Wistuba et al., 2016) is an ensemble approach where the Gaussian process surrogate of the target task is weighted with surrogates of the training datasets based on the ranking similarity of the evaluated hyperparameters;

7. **RGPE** (Feurer et al., 2018) is another ensemble approach, similar to TST-R, which estimates the weights by optimizing a ranking loss between the surrogates of the training datasets and that of the target task;

8. **ABLR** (Perrone et al., 2018) is a multi-task Bayesian linear regression approach that optimizes a shared feature extractor across the training datasets as an initialization strategy for the target task;

9. **GCP+Prior** (Salinas et al., 2020) utilizes a Gaussian Copula process (Wilson & Ghahramani, 2010) trained jointly on the training tasks, where a quantile-transformation is applied on their respective responses. The pretrained process is used as parametric prior for the target dataset;

10. **FSBO** (Wistuba & Grabocka, 2021) uses deep Kernel Gaussian processes (Patacchiola et al., 2020) to estimate the response of the target dataset. The parameters are initialized via meta-learning the joint response surface over the training datasets.

In a nutshell, our experimental protocol based on HPO-B is a large scale one by the standards of the prior papers, as it involves 10 baselines, 16 search spaces (algorithms whose hyperparameters we tune), 101 datasets, and totally 935 black-box tasks containing 6.3 million evaluations.

## 6.2 IMPLEMENTATION DETAILS

We implement the Deep Kernel Gaussian Process using GPyTorch 1.5 (Gardner et al., 2018) with a Matern 5/2 kernel. As described in Equation 4, DKLM is composed of two modules, $\phi := \phi_1 \circ \phi_2$. The parameters of the network have been selected based on the performance on a held-out validation set. Function $f$ is 4 dense layers with 32 hidden units and ReLU activation functions, whereas $g$ is 1 dense layer with 32 hidden units. Finally, $\phi_1$ is 4 dense layers with 32 hidden units and ReLU activation. All parameters of the Deep Kernel are estimated by maximizing the marginal likelihood. We achieve this by using gradient ascent and Adam (Kingma & Ba, 2015) with a learning rate of 0.001 and batch size of 64 with $t \in [2, 100]$.

## 6.3 HPO RESULTS AND DISCUSSION

We start by comparing **non-transfer** HPO methods to our deep GPs with meta-features DKLM (RI) that are randomly initialized (no meta-learning). As depicted in Figure 2, DKLM (RI) outperforms the baselines after 100 hyperparameter trials in terms of both the mean normalized regret and the mean rank metrics. We also notice how the performance improves gradually with the increasing number of trials, indicating the impact of the posterior variance modeling of our method (Section 5) as more observations are present on the black-box responses.

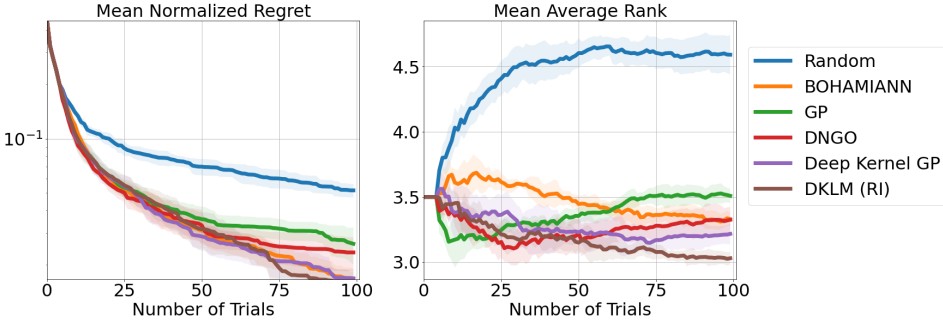

Figure 2: Aggregated comparisons of normalized regret and mean ranks across all search spaces for the non-transfer learning HPO methods on HPO-B-v3

Afterwards, we demonstrate the comparison of state-of-the-art **transfer** against our method in Figure 3. DKLM outperforms the rest of the baselines with lower mean normalized regret and lower mean rank. The superiority of landmark meta-features also becomes evident after a larger number of trials (more than 50) .

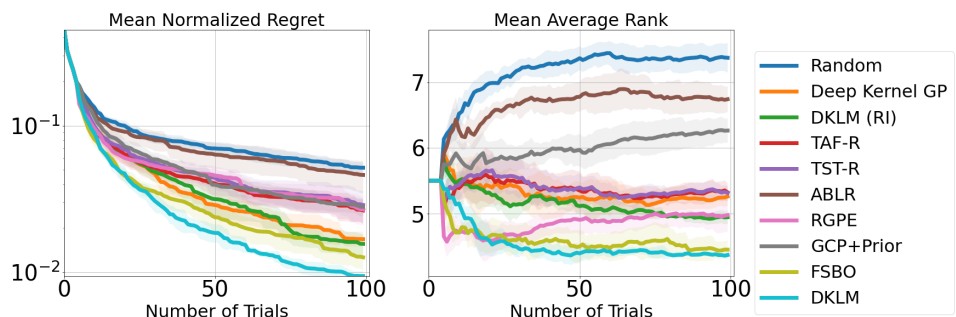

Figure 3: Aggregated comparisons of normalized regret and mean ranks across all search spaces for the transfer learning HPO methods on HPO-B-v3

To further inspect the results we show the performance of DKML and all other baselines in the selected individual search spaces of Figure 4. We notice primarily that meta-learning the initialization in DKLM improves the general performance in most cases. Nevertheless, we notice in 4796 that effect of transfer learning is not evident, as DKLM (RI) and Deep Kernel GP are better than the

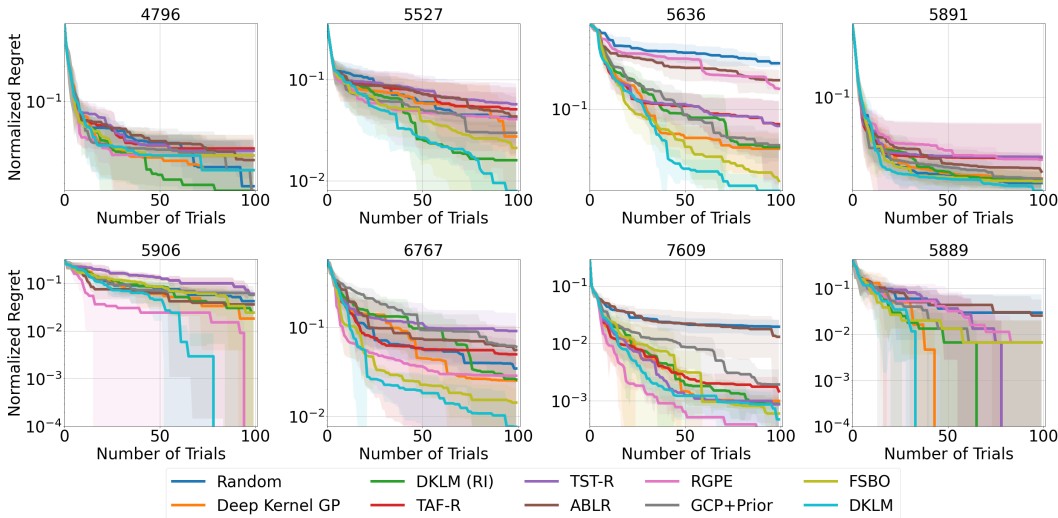

Figure 4: **Normalized regret** comparison of transfer learning HPO methods on selected benchmarks from HPO-B-v3

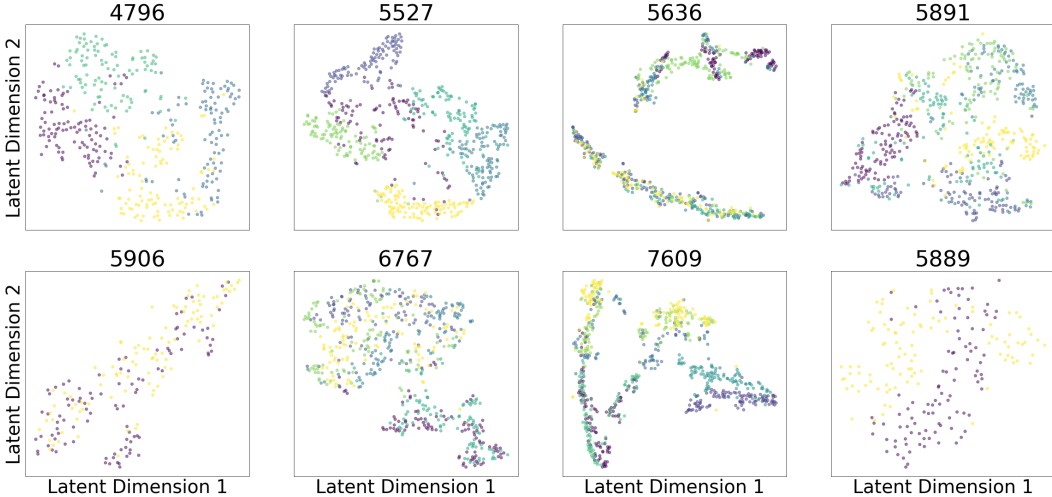

Figure 5: 2D illustration of meta-features extracted from each task in 8 selected search spaces. For each task, we sample 100 sets of 5 data points to extract landmark meta-features. We reduce the dimensionality of the meta-features into a 2D representation via TSNE (Liu et al., 2017).

meta-initialized variants, DKLM, and FSBO respectively. Still, landmark meta-features prove yet again that they are effective in a single task setting.

## 6.4 END-TO-END LANDMARK META-FEATURES

To motivate the importance of landmark meta-features, we illustrate in Figure 5 the 2D latent dimensions of the landmark meta-features for every test task in the 16 spaces of HPO-B-v3. Each point on the graph represents a set of meta-features extracted from 100 randomly sampled data points, i.e. $t = 100$, from the individual tasks after meta-initialization of the weights of DKLM. We observe that the same color-coded meta-features, i.e. belonging to the same task, lie generally within the vicinity of each other, and distant from other tasks. As pointed out by Jomaa et al. (2021a), any meta-feature extractor should be able to preserve inter-and intra-dataset similarity, a property that is evident here.

## 6.5 RESILIENCE TO NEGATIVE TRANSFER

To test the phenomenon of negative transfer we design a specific ablation experiment, where the datasets in the meta-test split are uncorrelated to the datasets in the meta-train split. First of all, we define the correlation of two datasets using Kendall's $\tau$ correlation coefficient (Kendall, 1945). We sample 100 random hyper-parameter configurations and query the validation accuracies of these configurations on both datasets from an HPO-B search space, to compute the Kendall's $\tau$ coefficient between the accuracies' vectors. Afterward, for every HPO-B search space, we i) compute the correlation between each meta-test dataset and all the meta-training datasets, ii) keep the 10% of the meta-training datasets that have the smallest mean Kendall's $\tau$ coefficient to the meta-test datasets.

We report the ablation results in Figure 6, where DKLM (NT) and FSBO (NT) denote methods trained using only the 10% most uncorrelated meta-training datasets. In contrast, DKML and FSBO are trained using all the 100% meta-training dataset. We notice that performance gap of the ranks between DKLM (NT) and FSBO (NT) is larger than that between DKLM and FSBO, validating DKML's resilience to the negative transfer phenomenon.

Additionally, we report the the critical difference diagrams @100 trials. We notice that the performance of DKLM (NT) is statistically significant compared to FSBO (NT), further illustrating the resilience of DKLM to negative transfer.

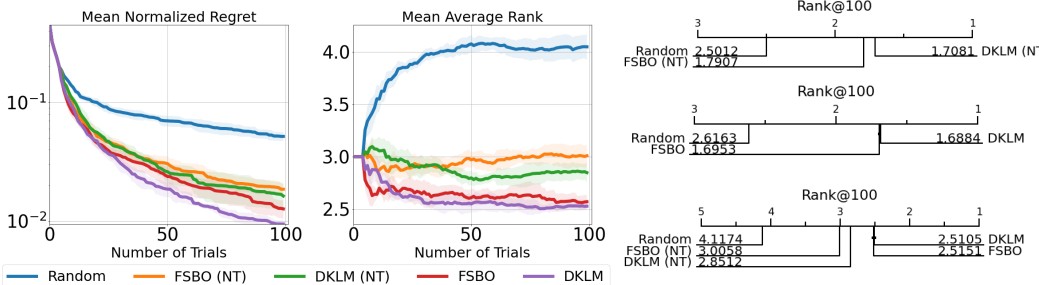

Figure 6: Aggregated comparisons of normalized regret and mean ranks across all HPO-B search spaces, where the gap between DKLM (NT) and FSBO (NT) demonstrates the resilience to negative transfer.

## 7 LIMITATIONS

Despite the fact that our method significantly reduces the time for fitting Machine Learning, we caution practitioners against overtuning their model for a large number of configuration trials, only to get a very small improvement in accuracy, unless it is absolutely necessary from a business need.

## 8 CONCLUSION

In this paper, however, we propose DKLM as a simple yet effective method to better condition deep kernel Gaussian Processes on tasks. Inspired by landmark meta-features, we design a set-based meta-feature extractor that captures the interaction between the available hyperparameters and their respective responses, and consequently generates distinct task-specific attributes. DKLM is meta-learned on a set of source tasks in an end-to-end fashion to jointly approximate the response surface over the shared hyperparameter and landmark meta-feature space. We show in a battery of experiments the significance of landmark meta-features, outperforming state-of-the-art HPO baselines in non-transfer and transfer learning settings.

## ETHICS STATEMENT

In our work, we use only publicly available data without privacy concerns. Furthermore, our algorithm reduces the overall time for fitting machine learning algorithms, therefore, saving computational resources and yielding a positive impact on energy consumption.

## REPRODUCIBILITY STATEMENT

We promote reproducibility as detailed below:

- We use only publicly available datasets.
- All our baselines are publicly available and provided by the HPO-B benchmark (Pineda-Arango et al., 2021).
- We clearly describe our method in Section 4 and provide implementation details in Section 6.2.
- Finally, we plan to make the source code of our method publicly available.

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

## A    ALGORITHMS

We describe in Algorithm 2 the general procedure for BO optimization. Our surrogate model, DKLM, is fit to the history of observations $\mathcal{H}^{t-1}$ at each trial via Algorithm 3.

---

**Algorithm 2:** Sequential Model-Based Optimization with DKLM

---

**Input:** kernel parameters $\theta$, network parameters $w$, black-box function $f$, $N$ initial evaluations $\mathcal{H}^0 = \{(x_i, y_i)\}_{i=0}^N$ on target task, acquisition function $\mathcal{A}(\cdot)$, number of trials $T$,
**Output:** Best configuration and response $(x_*, y_*)$

1 **for** $t = 1$ *to* $T$ **do**
2     Fit $\phi$ and kernel using $\mathcal{H}^{t-1}$ in Algorithm 3 ;
3     Obtain next configuration to evaluation $x = \arg\max_{x \in \mathcal{X}} \mathcal{A}(x)$ ;
4     Observe response $y = f(x)$;
5     Update History $\mathcal{H}^t = \mathcal{H}^{t-1} \bigcup \{(x, y)\}$ ;
6 **end**
7 Return best configuration $(x_*, y_*) = \arg\max_{(x,y) \sim \mathcal{H}^t} y$

---

---

**Algorithm 3:** Fitting DKLM to a target task

---

**Input:** Learning rates $\eta_1$ and $\eta_2$, $N$ evaluations $\mathcal{H} = \{(x_i, y_i)\}_{i=0}^N$ on target task, kernel parameters $\theta$, network parameters $w$
**Output:** Gaussian Process posterior $p(f_* | x_*, \mathcal{H})$

1 **while** *not converged* **do**
2     Compute marginal likelihood $\mathcal{L}$ on $\mathcal{H}$ (Equation 6);
3     $\theta \leftarrow \theta + \eta_1 \nabla_\theta \mathcal{L}$;
4     $w \leftarrow w + \eta_2 \nabla_w \mathcal{L}$;
5 **end**
6 Compute Gaussian Process posterior $p(f_* | x_*, \mathcal{H}) = \mathcal{N}(\mu, \sigma^2)$ (Equation 2)

---

## B    EXPERIMENTAL PROTOCOL

We followed the protocol presented in HPO-B, (Pineda-Arango et al., 2021), a benchmark collection of tasks collected from OpenML. The tasks are grouped in 16 search spaces, and every search space contains three splits: meta-training, meta-validation and meta-testing. According to the authors, the splitting for the percentage of tasks is 80%, 10% and 10%, respectively. We refer to the repository [1] to know the exact task IDs for every split.

In our experiments, we meta-train a model per search space, whereas we used the meta-validation data for early-stopping. We use $\eta = 0.001, v = 1, n = 1000, T_{\min} = 2, T_{\max} = 100, b = 64$ for meta-training (Algorithm 1). We optimize the black-box function following Sequential Model Based Optimization (SMBO, Algorithm 2), we use Expected Improvement as the acquisition function:

$$\mathcal{A}(x) = \mathbb{E}\left[\max\{f_*(x) - y_{\max}, 0\}\right] \tag{8}$$

which uses the mean and variance computations presented in section 3.2 for estimating the expectation. We fit DKLM using ADAM (Kingma & Ba, 2015) with learning rates $\eta_1 = \eta_2 = 0.001$, (see Algorithm 3). We use the meta-validation dataset for choosing the number of hidden layers in Equation 5 and the learning rates $\eta$.

## C    IMPACT OF SAMPLE SIZE

We mentioned in Section 6.5 that the performance of DKLM improves with the increasing number of observed data points. To further examine this observation, we present in Table 1 the negative log

---

[1]https://github.com/releaunifreiburg/HPO-B

likelihood given a history of size $t$. To get these results, we sample $t$ points for the history $\mathcal{H}^t$ and compute the negative log-likelihood on 100 separate data points upon meta-initialization of DKLM. The process is repeated 100 times for each task, for a total of 10000 data points. We highlight the trend of the negative log-likelihood by fitting a linear model to the 10000 points and reporting the slope. In most search spaces, the performance improves with the increasing number of observed data points, where we notice some over-fitting in others, e.g. space 5860 that can be alleviated by further regularization.

Table 1: Mean Negative Log-Likelihood upon meta-initialization of DKLM with 1 unit of standard deviation, lower is better.

| **Space** | $t = 5$ | $t = 10$ | $t = 20$ | slope |
|---|---|---|---|---|
| 4796 | $0.38 \pm 2.64$ | $-0.19 \pm 1.80$ | $-0.78 \pm 0.99$ | -0.075 |
| 5527 | $2.92 \pm 6.12$ | $1.22 \pm 3.86$ | $0.9 \pm 2.44$ | -0.12 |
| 5636 | $-1.8 \pm 3.96$ | $-3.01 \pm 0.72$ | $-3.11 \pm 0.55$ | -0.077 |
| 5859 | $22.78 \pm 40.70$ | $10.18 \pm 28.51$ | $-0.37 \pm 10.09$ | -1.473 |
| 5860 | $12.76 \pm 21.81$ | $14.61 \pm 19.42$ | $16.13 \pm 18.1$ | 0.215 |
| 5889 | $0.22 \pm 1.27$ | $1.84 \pm 1.86$ | $3.75 \pm 2.97$ | -0.039 |
| 5891 | $0.33 \pm 0.90$ | $0.02 \pm 0.86$ | $-0.28 \pm 0.69$ | 0.414 |
| 5906 | $2.91 \pm 4.04$ | $5.52 \pm 4.17$ | $9.22 \pm 5.19$ | -0.003 |
| 5965 | $-0.62 \pm 1.12$ | $-0.93 \pm 0.75$ | $-0.73 \pm 0.77$ | -0.015 |
| 5970 | $-0.99 \pm 0.12$ | $-1.11 \pm 0.10$ | $-1.22 \pm 0.09$ | 0.021 |
| 5971 | $-1.08 \pm 0.73$ | $-1.26 \pm 0.77$ | $-0.81 \pm 1.10$ | 0.14 |
| 6766 | $0.29 \pm 0.31$ | $0.95 \pm 0.43$ | $2.38 \pm 0.59$ | -0.118 |
| 6767 | $1.59 \pm 5.64$ | $-0.05 \pm 1.43$ | $-0.39 \pm 0.92$ | -0.032 |
| 6794 | $-0.54 \pm 1.76$ | $-1.18 \pm 0.82$ | $-1.12 \pm 0.67$ | 0.019 |
| 7609 | $-0.59 \pm 2.42$ | $-1.49 \pm 1.74$ | $-1.89 \pm 0.61$ | -0.08 |
| 7607 | $-0.05 \pm 0.77$ | $-0.06 \pm 0.87$ | $0.21 \pm 0.76$ | 0.229 |
| **Average** | $2.41 \pm 6.40$ | $1.57 \pm 4.69$ | $1.37 \pm 4.86$ | -0.062 |

## D    DETAILED EXPERIMENTAL RESULTS

We present the detailed results for the 16 search spaces in Figure 7. Additionally, we plot the 2D landmark meta-features in Figure 8.

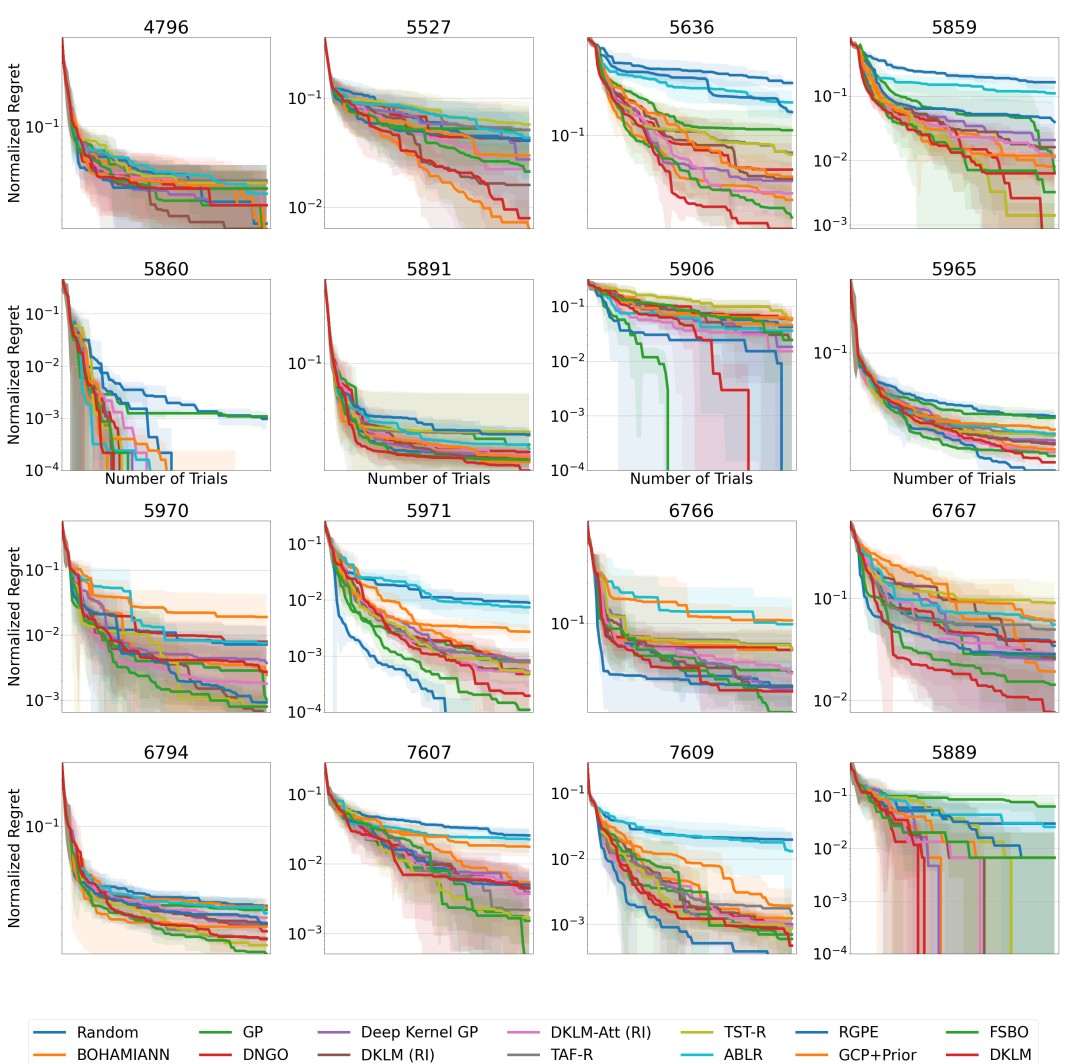

Figure 7: **Normalized regret** comparison of transfer learning HPO methods on all benchmarks from HPO-B-v3

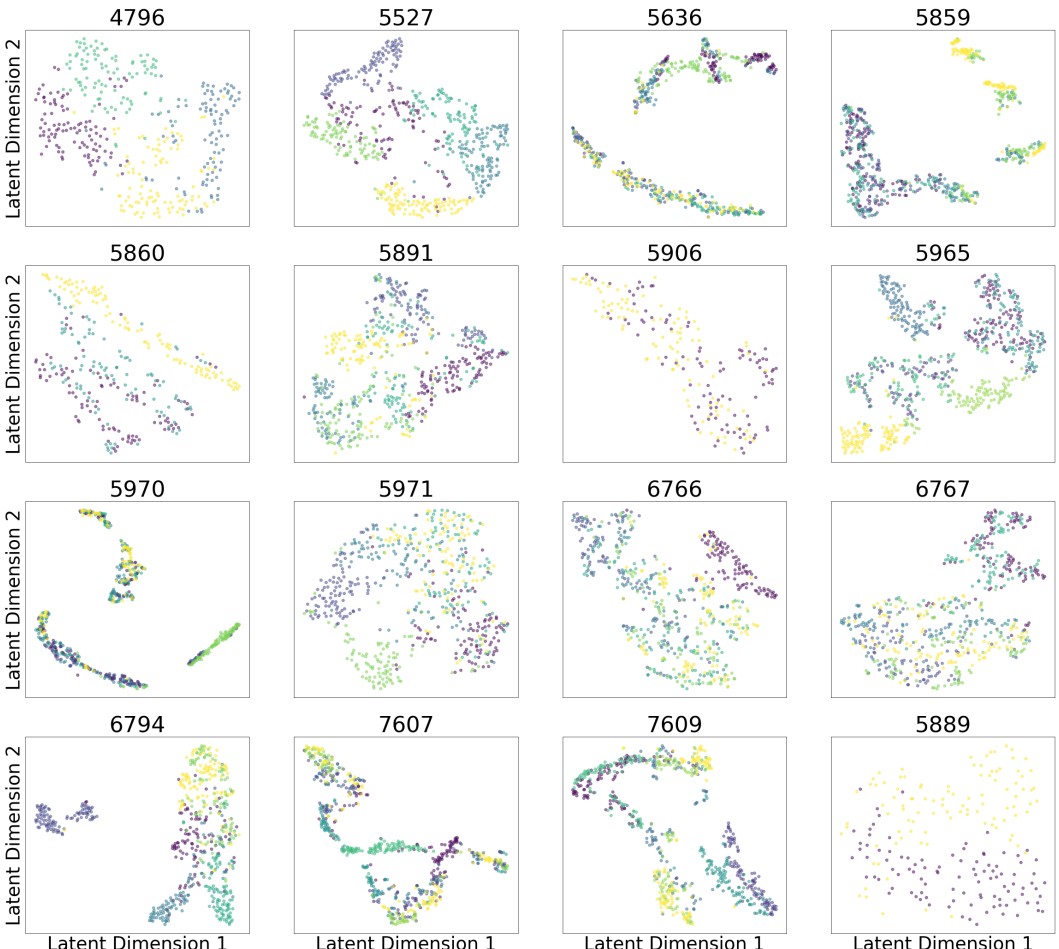

Figure 8: 2D illustration of meta-features extracted from each task in the 16 selected search spaces. For each task, we sample 100 sets of 5 data points to extract landmark meta-features. We reduce the dimensionality of the meta-features into a 2D representation via TSNE (Liu et al., 2017).

