# OpenReview forum: "Transfer Learning for Bayesian HPO with End-to-End Meta-Features"
_ICLR.cc/2022/Conference — ICLR 2022 Submitted_

### Official Review · Reviewer_1rkp · 2021-11-02

**Correctness:** 3
**Technical Novelty And Significance:** 3
**Empirical Novelty And Significance:** 3
**Recommendation:** 5
**Confidence:** 3

**Main Review:**

## Strengths

I believe that there are merits to the proposed DKLM approach.
It makes intuitive sense to me that the Deepset style inclusion of prior observations is preferable to using prior observations as initialisation for the GP as done by prior work.

The hpo-b-v3 experiments look somewhat convincing.
DKLM performs decently in the "non-transfer setting" and impressively well in the transfer setting.
I'm inclined to believe the validity of these results, although I wish the authors would have included code / more details of the setup.

## Weaknesses

Unfortunately, I feel that many parts of this paper are severely lacking in clarity.
The writing often feels unnecessarily convoluted and I am sometimes unsure of what exactly the authors are trying to tell me with a particular paragraph.
Further, the lack of clarity unfortunately leads to open questions about the proposed approach itself.
Below I give specific details of this.
I hope the authors understand my criticisms as constructive and take them as an opportunity to improve the current draft of the paper.

### Introduction

By the end of the introduction, I neither understand what open challenges in HPO are, nor what DKLM does differently to established works.
Concretely, I find the following sentences in the introduction more confusing than helpful:
* "we propose a novel architecture for deep GP kernels (Wilson et al., 2016a) that are enriched with novel end-to-end neural network components that generate meta-features only from the tuples of past hyperparameter configurations and their evaluated performances" --> How do I enrich a deep GP kernel with neural network components? Don't Deep Kernel GPs already have neural networks do produce the encoding?
* "Introduce the first paper that tackles the negative transfer phenomenon in Bayesian HPO" → What is the negative transfer phenomenon? This is not explained here.
* "Propose an end-to-end deep GP which implicitly learns networks that generate metafeatures," → What does it mean when a GP "implicitly learns networks"? Are you referring to the neural network of the deep kernel? Why does this learn this only "implicitly"?
In contrast,  Jomaa et al. (2021a) and Wistuba and Grabocka (2021), which might be the most related work to this submission, are much much more approachable.

### Negative Transfer

This paper often uses terms without introducing them.
For example, one of the contributions listed in the introduction is that this paper is "the first paper that tackles negative transfer phenomenon in Bayesian HPO".  They do not explain anywhere what the 'negative transfer phenomenon is'. The authors do give a citation but I do think it would make sense to briefly explain this phenomenon to make the paper more accessible to readers.
(The negative transfer paper (Wang et al., 2019) itself gives a great summary in the first two sentences of the papers abstract.)
However, it is not clear to me how exactly DKLM tackles negative transfer in ways that prior transfer learning HPO works have not?
(The following sentence of the paper does *not* answer my questions: "By adding the task-specific information of the meta-features, the GP surrogate can infer a more accurate response surface on a new task based on similar source tasks that share similar meta-features. Therefore, our method is the first to tackle the negative transfer phenomenon for Bayesian HPO.")
The negative transfer problem is also not once mentioned in the experimental evaluation.

### Landmark  Features / Meta-Features

The authors repeatedly claim that DKLM (**D**eep **K**ernel Gaussian Process surrogate with **L**andmark **M**eta-features) makes use of "landmark meta-features" (Pfahringer et al., 2000; Feurer et al., 2014).

In prior work, landmark features refer to features that characterise a dataset via the predictive performance obtained by a variety of simple models on that dataset.
The idea here is that the performance of different machine learning models can be used to characterise the dataset itself: dataset A works well with tree-based models, Dataset B works well with simple parametric models, and so on.
This information can then be used as features in HPO (Feurer et al., 2014).
Note that, here, these cheap models are not the model for which we want to perform HPO, but instead are additional models that are only used to characterise datasets.

This is, to my understanding, the definition of landmark features.
In this paper, no definition is given, except that landmark features are " typically estimated by measuring the response of given datasets to machine learning algorithms".
It is entirely unclear to me, how anything that DKLM does can be called "landmark features" given that no simple auxiliary models are trained whose performance is used as a dataset-specific feature.
Again, maybe this misunderstanding could have been resolved if the authors had discussed these terms in more detail.

Additionally, the paper claims that DKLM learns to generates *meta-features* which it defines as "capturing the similarity between datasets".
However, DKLM learns embeddings that depend only on hyperparameters and loss values.
No dataset-specific information is available for constructing these embeddings.
The argument of the paper here seems to be that the embedding network learns a representation that learns to 'cluster' different dataset types.
However, given that no information of the dataset is available to the embedding, this seems unnecessarily complex/indirect.
The experiments of Figure 5 (weakly) support the idea that this clustering does occur.
(Also why not apply the approach of Jomaa et al. to actually include learned dataset-specific features? If DKLM is confronted with a novel dataset, unlike the approach of Jomaa et al., it cannot make any judgement of it before observing loss-response pairs.)
In any case, I feel that this would need to be exposed much more clearly.

In contrast, the approach of Jomaa et al. (2021a) also claims to learn meta-features.
They construct a neural network architecture that take as input a literal dataset (with all rows/datapoitns and columns/features) and outputs a learned representation.
This per-dataset representation is then useful for HPO.
The definition of Jomaa et al. (2021a) for learning meta-features makes sense to me.

What are the definitions of landmark features/landmark meta-features/meta-features used in this paper and how are they consistent with those of earlier works?

### Missing DKLM Details

The exposition of the approach is lacking both clarity and detail.
In that regard, equations (4-7) are extremely helpful in understanding what's going on.
Without them, I'm not sure if I would have any idea of what happens in DKLM.

However, the exposition of DKLM is, in my eyes, not complete and lacking many details.
For example, this is a list of questions that I still have about how exactly DKLM works:
* Algorithm 1 gives a recipe for meta-learning with DKLM. But what does applying DKLM look like? Do you continue to train $w$ or $\theta$ when applying DKLM to a HPO task? (E.g. is anything retrained for the acquisition steps of Figure 3?) Maybe an Algorithm 2 illustrating the application of DKLM would be helpful here.
* Also, when applying DKLM to a dataset, you update the history $\mathcal{H}_t$ with observations from the current dataset, right? If so, what happens if you don't do this? You would still add new observations to the GP (just not use them in $\phi_2$).
* However if you don't do update $\mathcal{H}_t$ as you apply DKLM to a dataset,  what does the "DKLM *without* transfer learning" setting of Figure 2 look like? This is not explained in the main body of the paper.
* What acquisition strategy do you use for HPO? Expected improvement? As far as I can tell this is not stated in the paper.
* From Algorithm 1, I can see that you are using REPTILE to train DKLM. Why are you using REPTILE? This is not discussed in the main text at all.
* The information in 6.2 is not enough to replicate the experiments. How do you set the deep kernel GP hyperparameters? What are the training schedules? Do you optimise the GP hypers? It would be great if an appendix with this information and/or code of DKLM were provided by the authors.

### Experimental Results

For the toy experiments, it seems to me that DKLM should be able to predict close to perfect after having found a single peak value of the sinusoid in the transfer learning setting.
After all, a single peak value completely defines the function. (Technically speaking, obtaining a single derivative of the function, e.g. approximated by two close observations, should also be sufficient.)
However, as is evident from Fig. 1 (top left), the predicted function is far from perfect despite observations near the peak.
Based on my understanding, DKLM should be able to learn to perform similarly to a maximum likelihood fit of  $a * \sin(x)$ to the observations.
However, the results displayed here are looking far worse.
Weirdly, going from DKLM + 2 trials to DKLM + 3 trials, the uncertainty of the predictions *increases* despite this not making much intuitive sense.


## Smaller Comments

* "Nevertheless, extracting meta-features requires direct access to the datasets, which might be difficult in real settings where only the meta-dataset is available. " Why is direct access to datasets not available in real settings? I thought a meta-dataset directly *contained* datasets (and gives direct access to them)? Unfortunately the term meta-dataset is not further defined in the context of this paper.
* S.3.2: Why is the noise variable $\sigma_n$ indexed with $n$?
* "We also notice how the performance improves gradually with the increasing number of trials, indicating the impact of the posterior variance modeling of our method (Section 5) as more observations are present on the black-box responses."
    * It's unclear to me why "posterior variance modeling" is needed for gradually increasing performance? Also, other HPO approaches also rely on GPs with posterior variances. What is unique about DKLM here?

## Language & Style

I believe the writing style of this paper needs improvement. Below, I present some specific examples and suggestions of where I think the writing could be improved.

* "however, it [HPO] remains an open problem" --> Why is HPO an open problem? When would it no longer be an open problem? I feel like saying that HPO is an open problem is either trivial or not true, and so best avoided.
* "In contrast to existing approaches, we propose a novel Deep Kernel Gaussian Process surrogate with Landmark Meta-features (DKLM) that can be jointly meta-trained on a set of source tasks and then transferred efficiently on a new (unseen) target task." --> Why is your work *in contrast* to existing approaches? It seems that you, *like* prior work, apply transfer learning to HPO.  It seems to me that "jointly meta-training on a set of source tasks" is also not novel.
* "prior work focus on" → focuses
* "Transfer learning for HPO has been observed by modeling tasks jointly" → maybe "has been applied". "Observed" makes it sound a bit like an event in nature.
* "Among *existing* ~~the variety of~~ acquisition functions, ~~the~~ expected improvement is widely adopted (Mockus, 1974)."
* "The standard approach of fitting GPs is to optimize the weights of the kernel function, e.g. squared exponential kernel, θ."  I think it is misleading to call the optimisation of the kernel *hyperparameters*  'fitting GPs'. Gaussian Processes perform Bayesian *inference* and are not "fit" by "optimising" any parameters. However, the hyperparameters in the GP (but not the GP itself) are often inferred by maximising the evidence.
* "Consequently, the solution of the joint model resides on a local minimum so that given limited information about the new (unseen) target task [______] " This sentence is missing an ending? What happens given limited information about the task?


**Summary Of The Paper:**

This paper introduces DKLM, a novel approach to hyperparameter optimisation based on transfer learning.
DKLM uses a Deep Kernel GP (Wilson et al., 2016) surrogate to predict test losses / response functions for given hyperparameter configurations.
The embeddings of the Deep Kernel GP are obtained by aggregating over hyperparameter-response pairs of previously observed datasets.
This is the main mechanism through which the HPO is conditioned on previously observed experiments.
The embedding mechanism relies on a Deepset (Zaheer et al., 2017) style neural network encoding, that sums over all hyperparameter-response pairs.
Experiments demonstrate DKLM improves performance over prior work.

**Summary Of The Review:**

The proposed method, DKLM, for transfer-learned HPO seems interesting and is, as far as I'm aware, novel.
The experimental evaluation seems to demonstrate benefits of DKLM over prior work.
Unfortunately, the paper is (sometimes severly) lacking in clarity, and the draft requires significant further work before being, in my opinion, acceptable for publication.
The exposition leaves me with too many open questions as to how DKLM actually, as well as some of the experiments, actually work.
Further, one of the main contributions of the paper – avoiding 'negative transfer' – is not discussed sufficiently and not evaluated at all, and I have trouble following the authors in why it is appropriate to say that DKLM learns "landmark meta-features".
If the authors can address these main points of my review, I am happy to reconsider my score.

Thank you for including reproducibility and ethics statement. Please make code for reproducing experiments available as soon as possible.

---

> ### Author Response · Authors · 2021-11-17
> **Rebuttal**
>
> Thank you for your insightful comments.
> # Introduction
> - The deep GP uses a deep kernel for capturing the similarity of a pair of hyper-parameter configuration vectors. However, it does not capture the similarity between datasets. Because the characteristics/meta-features of a dataset do not depend only on a pair of configurations but on the full set of configurations and their evaluated response. Such information is not directly inherent to the standard deep GP.
>
> - Negative transfer implies a poor generalization performance on test tasks that are dissimilar to the training tasks, according to a predefined dissimilarity measure, e.g. similarity of response curves. We added this description to the introduction.
>
> - We additionally added an additional experiment that compares FSBO and DKLM when trained on uncorrelated source tasks, Section 6.5.
>
> - The network implicitly learns to generate task-specific representations that we refer to as land-mark meta-features. It learns them implicitly because the overall objective is maximizing the marginal loglikelihood of the surrogate in an end-to-end manner, unlike how Jomaa 2021a explicitly did via a batch identification loss.
>
> # Landmark Meta-features
>
> - You are entirely correct with regards to the definition of landmark meta-features. We provide a generalized way of learning landmark-meta features.
> - Considering the classical formulation: we have K classifiers f_1,..,f_K, each achieving a performance P on a dataset as P_k(D). In essence, concatenate the classification accuracies of the classifiers and we get a K-dimensional landmark meta-feature vector [P_1(D), ..., P_K(D)].
> - Notice that what we have achieved is given D compute a "function with a K-dimensional output" (a.k.a. performance of K classifiers given D). As a result, we can easily consider the end-to-end approximation [P_1(D), ..., P_K(D)] = f(D), where f: D -> R^K with f being a neural network defined on a dataset D.
> - In our case, we do not approximate the performance of classical classifiers with our function f (although that is feasible given the universal approximation nature of neural networks). Instead, we compute f more optimally by a meta-feature network for helping the actual HPO loss function we are interested in in an end-to-end manner.
>
> # Missing DKLM Details
>
> - DKLM is meta-initialized on the target task after meta-learning on the source tasks. We follow the same protocol as any Bayesian Optimization solution, Algorithm 2 in the Appendix. We iteratively select new configurations by maximizing an acquisition function, in this case Expected Improvement, as described by the experimental protocol, Appendix B.
>
> -  DKLM without transfer learning is the variation of the model that is randomly initialized, and not meta-initialized. This is explained in Section 6.3.
>
> - We implemented	REPTILE because it has empirically shown that it performs well in several settings, FSBO, Jomaa et al 2021b and because it is faster than MAML due to not needing a double-derivative update rule.
>
> - In addition to 6.2, we added the more details in Appendix B and uploaded the source code.
>
> # Language
> We have taken some of your comments into consideration in the updated draft.
>
> We hope that these answers are sufficient to mitigate any concerns. Please feel free to raise any other issue that might prevent you from improving your score.

---

> > ### Comment · Reviewer_1rkp · 2021-11-20
> > **Reviewer Response**
> >
> > Thank you for the detailed reply to my rebuttal!
> >
> > In particular, thank you for including an experiment on the negative transfer phenomenon, for including Algorithms 2 and 3, for clarifying the acquisition function used, and thanks for taking some of my comments on language/missing details into account.
> >
> > ## Further Comments
> >
> > I'm not sure I share your interpretation of the new negative transfer results. It seems to me that both DKLM and FSBO take a decided performance hit when being meta-trained on uncorrelated datasets.
> > You write that the "performance gap of the ranks between DKLM (NT) and FSBO (NT) is larger than that between DKLM and FSBO" but wouldn't be how much of a relative performance decrease DKLM and FSBO have? Or how far their rank decreases for this test? (This would require more than two methods though.) As far as I can tell both methods take a decided hit in performance (if anything the difference between DKLM (NT) and DKLM is larger than the difference between FSBO (NT) and FSBO).
> >
> > > we can easily consider the end-to-end approximation [P_1(D), ..., P_K(D)] = f(D), where f: D -> R^K with f being a neural network defined on a dataset D
> >
> > If I'm not mistaken, you never take a dataset as input though. Further, it is unclear if your network actually learns to "approximate the performance of classical classifiers with our function f".  I am not yet convinced it makes sense to 'lump together' different datasets in the encoding, without giving the model the 'explicit' option to learn a dataset representation.
> >
> >
> > ## Bottom Line
> >
> > Many of my concerns regarding the clarity of this submission have not been addressed.
> > I am not convinced of the interpretation that the authors have of the negative transfer experiment.
> >
> > The confusing use of the term 'landmark meta-features' has not been addressed.
> > Compared to prior work, I still find this publication hard to read.
> >
> > I feel that some detail is still missing form the current draft:
> > * My concerns on the toy experiments have not been addressed.
> > * It is still unclear to me why DKLM should be suited to addressing negative transfer.
> > * A discussion of why you use REPTILE should be included in the paper.
> > * What is the shading in Figure 6?
> > * Algorithm 3 suggests you use full-batch gradient descent to fit the model. However, the text mentions ADAM being used on mini-batches.
> > * How is DKLM applied in a non-transfer setting?

---

> > > ### Author Response · Authors · 2021-11-23
> > > **Reply**
> > >
> > > ## Regarding Negative Transfer
> > > We added to Section 6.5 the critical difference diagram between FSBO and DKLM in negative transfer and standard setting. In a negative transfer setting, the performance of DKLM (NT) is statistically significant compared to FSBO (NT). We would like to point out that the hit in performance is due two-fold, learning on uncorrelated tasks, as well as having less source tasks for meta-training. We hope that this is sufficient to prove that DKLM is robust against negative transfer compared to state-of-the-art.
> > >
> > > ## Regarding Concerns
> > >
> > > - DKLM is suited for negative transfer because it forces the surrogate Deep Kernel GP to be conditioned on learnable task-specific meta-features, generated from the hyperparameter configurations and their responses.
> > > - Shading in Figure 6 is the standard deviation across search spaces
> > > - Toy experiment: The function is **not defined by the peak alone** but as we point out, we meta-train on a collection of sinusoid functions of the form $f^{(k)}(x) = a^{(k)}\sin(b^{(k)}x)$. Only one measurement is not sufficient to perfectly approximate a sine wave, because, there could be multiple sine waves passing through that point. How do you know that an observed point is "the peak" of the sine wave?
> > >
> > > - We use the full batch of data points during fitting of the target surrogate during inference, whereas we use Adam for meta-training, Algorithm 1. For simplicity in Algorithm 1, lines 11 and 12 show a gradient descent update
> > > - In a non-transfer setting, DKLM is initialized randomly, instead of via meta-learning the initialization, and is referred to as DKLM (RI)
> > > - REPTILE: We did not perform an ablation to highlight why REPTILE was chosen, however, based on the literature and state-of-the-art, it has been shown that REPTILE outperforms MAML, [1,2] and is faster as we do not need to compute a second-order derivative.
> > >
> > > References:
> > >
> > > - [1] Wistuba, Martin, and Josif Grabocka. "Few-shot bayesian optimization with deep kernel surrogates." arXiv preprint arXiv:2101.07667 (2021).
> > > - [2]  Jomaa, Hadi S., Lars Schmidt-Thieme, and Josif Grabocka. "Hyperparameter Optimization with Differentiable Metafeatures." arXiv preprint arXiv:2102.03776 (2021).

---

> > > > ### Comment · Reviewer_1rkp · 2021-11-24
> > > > **Reply**
> > > >
> > > > Thanks a lot for your reply and thanks for the clarifications! I will take these into account.

---

> > > > ### Comment · Reviewer_1rkp · 2021-11-28
> > > > **Response**
> > > >
> > > > Thanks again for your response.
> > > >
> > > > As I've mentioned in my original review, the experimental results do seem convincing, and I appreciate the detail added during the rebuttal.
> > > >
> > > > However, I have to say that I still feel that this submission is lacking in clarity of presentation and does not give sufficient intuition into why DKLM is a good/better thing to do than prior work. For example, see below for some points I still don't think have been addressed sufficiently.
> > > >
> > > > > Only one measurement is not sufficient to perfectly approximate a sine wave, because, there could be multiple sine waves passing through that point.
> > > >
> > > > In the paper, you show predictions after up to 8 observations. It's unclear to me why DKLM should have uncertainty about the predictions left at this point, given sufficient training data. (A naive fit of the observations to $a \sin(x + b)$ should be pretty good by then.)
> > > >
> > > > > DKLM is suited for negative transfer because it forces the surrogate Deep Kernel GP to be conditioned on learnable task-specific meta-features, generated from the hyperparameter configurations and their responses.
> > > >
> > > > I am not happy with this answer.
> > > > Unlike approaches that explicitly include dataset specific features, DKLM has no way of knowing that any test-time dataset is going to be dissimilar to one of the training datasets. All dataset-similarity has to be implicitly encoded in the learned deep kernel embeddings and then iteratively discovered over multiple trials at test time. How is this an architecture that's going to perform well for negative transfer? Why would this perform better than FSBO? If negative transfer is one of the motivations for DKLM, I would hope to see further discussion on it.

---

### Official Review · Reviewer_5B4g · 2021-11-02

**Correctness:** 4
**Technical Novelty And Significance:** 2
**Empirical Novelty And Significance:** 2
**Recommendation:** 6
**Confidence:** 3

**Main Review:**

Strengths
-------------
- This paper combines two intuitive ideas, and the method is fairly easy to understand
- Experimentally, aggregated metrics showcase the strength of the proposed DKLM method compared to other baselines
- The ablation test which removes the _learned_ meta features convincingly shows that investing additional effort into obtaining trained meta-features improves performance.

Weaknesses
-----------------
- The major weakness of this paper is that it seems to simply combine to well-known ideas (deep sets + deep kernel GPs), and as such has limited novelty.
- The clarify of the paper could be improved. In particular, the experimental setup was not entirely obvious to me. Is the optimization done as Bayesian optimization (with a DKLM in the Gaussian process)? If so, which acquisition function was used?

Comments/questions
----------------------------
- You mention the "negative transfer phenomenon" several times as a major motivation for this work; I would recommend describing it in more detail in the paper.
- I'm curious about the performance variability of DKLM: do you have any insight into which types of datasets or experimental settings DKLM tends to perform better or worse on, compared to other baselines?
- Did you compare DKLM to a GP parameterized with the learned metafeatures from Dataset2Vec?


**Summary Of The Paper:**

This paper focuses on doing transfer learning from related tasks for the optimization of ML model hyperparameters.

Previous work (e.g., Vanschoren, 2018) has considered meta-features as representation of related datasets or tasks; these features are incorporated into the model when transferring to the task at hand. This work proposes to _learn_ these meta features using a deep-set representation (Zaheer et al., 2017). These meta features are then passed to a kernel function that, finally, parameterizes the Gaussian process used during Bayesian optimization.

Experimentally, the authors compare their method to several baselines, both in the transfer and non-transfer learning settings; aggregated metrics show that the proposed method (DKLM) outperforms other baselines on average.



**Summary Of The Review:**

This is an interesting paper, but its novelty is limited. A more detailed analysis of the proposed method (in particular, ablations that use a deep kernel GP with different learned meta-features) would increase the impact of this work.

---

> ### Comment · Area_Chair_5eyF · 2021-11-10
> **elaborate a bit more?**
>
> Reviewer,
> Could you remark a bit on your opinion of the evidence in the submission supporting the claim in the abstract that the work "demonstrate[s] the empirical superiority of our method against a series of state-of-the-art baselines"? In particular, suppose for the sake of argument that we assume the work presents a "straightforward combination of well-known ideas", we might still imagine accepting such a manuscript if the work made a strong claim to perfecting the method, reduced it to practice uniquely effectively, or got particularly impressive empirical results. For a submission to be accepted using this line of reasoning, it would be crucial for the empirical results to be of some practical interest and impressive in some way, or for the experimental data *by itself* to be valuable enough in terms of insights, etc.
>
> Specifically, it would help if you could share your thoughts on the following questions:
> 1. How "realistic" or practically interesting are the tasks the paper presents results on?
> 2. Qualitatively, how strong are the results?
> 3. What is your assessment of the strength of the baselines?
> 4. How valuable are the experimental details?
>
> To be clear, at this point I as AC am not taking a position on the answer to these questions.

---

> > ### Comment · Reviewer_5B4g · 2021-11-24
> > **Empirical evaluation**
> >
> > Thanks for the questions!
> >
> > To clarify, as mentioned above to the authors, my opinion on the novelty was based mostly on the perspective of combining deep sets with a deep kernel GP, rather than, for example, using on surrogate features to mitigate the negative transfer phenomenon.
> >
> > 1. How "realistic" or practically interesting are the tasks the paper presents results on?
> > The dataset HBO-B used [1] is a super-set of tasks that are typically used to evaluate transfer learning methods, with unified preprocessing. Although I would prefer if this dataset were less recent and had been used by more previous work to further justify it as the evaluation benchmark, it appears to be both comprehensive and appropriate for this paper. It is worth noting, though, that HBO-B _lacks_ evaluations of SOTA deep learning methods, which are one of the highest consumers of HPO.
> >
> > 2. Qualitatively, how strong are the results?
> >   - The results are strong: the proposed method (DKLM) performs on average better than the considered baselines (Figure 3). When DKLM is not the best method, it is remains within the top methods (Figure 4). The ablation test on the value of using learned (rather than randomly initialized) meta features further justifies the method design.
> >
> >   - If this paper is considered as a purely empirical paper (which I am not sure it should be, as mentioned above), I think the empirical evaluation is not sufficient for acceptance. In such a case, I would have liked to see a more in-depth analysis of the non-aggregated performance of DKLM. I asked in my original review "do you have any insight into which types of datasets or experimental settings DKLM tends to perform better or worse on, compared to other baselines?", and I think this question should be answered for a purely experimental paper. Unless DKLM is systematically within the top-$n$ methods (let's say $n=3$ for the purpose of this argument), this paper would be made stronger by a detailed discussion of _when_ and _why_ DKLM can fail. If DKLM is always within the top-3 methods, this should be mentioned explicitly (this does not appear to be the case from Figure 7).
> >
> > - As pointed out in my initial review, I also think that other ablation tests would strengthen the paper, for example comparing the learned meta features to expert-created meta-features.
> >
> > - A discussion of the sensitivity of the method to the hyperparameter choices (in particular of functions $f$ and $g$) would also be welcome.
> >
> > 3. What is your assessment of the strength of the baselines?
> > The paper includes 10 baselines and covers both transfer and non-transfer learning methods. Some transfer learning methods closer to RL (e.g., Witsuba et al., 2018; Volpp et al., 2020) were not included and would have been nice to have, but given that those methods focus on the acquisition function rather than the surrogate model, it seems reasonable to not have included them. As such, I find that the baselines are quite strong.
> >
> > 4. How valuable are the experimental details?
> > The experimental details should be enough to reimplement and analyze DKLM.
> >
> > To summarize: I do not believe this paper should necessarily be viewed entirely as an experimental contribution, as was pointed out by the authors as a reply to my review. If it were to be evaluated in such a way, however, I think the analysis would need to be more in-depth: more ablation tests, further analysis of the failure modes instead of aggregated or punctual examples (for example: provide the datasets where DKLM did the worst, the best, and explain qualitatively what caused this gap).
> >
> > [1] HPO-B: A Large-Scale Reproducible Benchmark for Black-Box HPO based on OpenML

---

> ### Author Response · Authors · 2021-11-17
> **Rebuttal**
>
>
> - We kindly have a different view on the novelty aspect. To the best of our knowledge, ours is a novel method in two directions: it is both the first paper that conditions HPO surrogates on meta-features, as well as the first paper that actually learns meta-features in an end-to-end manner. We invite the author to point out published prior works that also conditions surrogates on meta-features in the context of Bayesian Optimization.
>
> - We follow the same experimental protocol proposed by HPO-B to allow for a fair comparison between the published results and our own. We have added the details to the Appendix B.
>
> - We propose DKLM as a new surrogate to replace the standard Gaussian process conventionally used in Bayesian optimization. DKLM is first learned to jointly approximate the response surface of the source tasks, Algorithm 1, and then fine-tuned to the target observations through a sequential model-based optimization, Algorithm 2 in the Appendix A. This is consistent with all the baselines that are iteratively fine-tuned to the observations of the target task.
>
> - You are right, the acquisition function is missing. We use expected improvement as the acquisition function and point that out in Appendix B.
>
> - Negative transfer implies a poor generalization performance on test tasks that are dissimilar to the training tasks, according to a predefined dissimilarity measure, e.g. similarity of response curves. We added this description to the introduction.
>
> - We additionally added an experiment in Section 6.5 that compares FSBO and DKLM when trained on **uncorrelated** source tasks in order to test our method’s resilience to negative transfer.
>
> - **Regarding Dataset2Vec** This is a very nice observation. Unfortunately, incorporating meta-features extracted by dataset2vec is not possible for this benchmark. Namely, because the primary datasets, i.e the dataset for which the algorithms in the meta-datasets were tuned are not available in HPO-B. We only have access to the response and the configurations. HPO-B does not provide the raw dataset, attributes and classes, by default.
>
> Please feel free to let us know if any more concerns should still arise that would prevent you from raising your score toward acceptance.
>
>
> References:
>
> HPO-B: Sebastian Pineda-Arango, Hadi S. Jomaa, Martin Wistuba, and Josif Grabocka. HPO-B: A largescale reproducible benchmark for black-box HPO based on openml. Accepted at NeurIPS Datasets and Benchmark Track, 2021.

---

> > ### Comment · Reviewer_5B4g · 2021-11-24
> > **Re: novelty & additional experiments**
> >
> > I thank the authors for their reply to my review.
> >
> > The additional experiments on the negative transfer phenomenon are very welcome and, from my perspective, increase the impact of this work. Regarding the novelty: my perspective is that of someone more familiar with the deepset literature than that of transfer learning, and with that in mind I recognize I may have missed some context in that regard.
> >
> > I understand why the ablation experiment I was thinking of wrt. Dataset2Vec is not possible. Nonetheless, I believe an investigation of the impact of learned vs ad-hoc meta features would strengthen the experimental section of this work.

---

### Official Review · Reviewer_kJb9 · 2021-11-02

**Correctness:** 4
**Technical Novelty And Significance:** 3
**Empirical Novelty And Significance:** 3
**Recommendation:** 8
**Confidence:** 3

**Main Review:**


Strengths
========
The paper very clearly explains the background, existing methods, and issues. It shows how various existing techniques and ideas (deep kernel Gaussian processes, landmark meta-features, deepset) are combined with meta-learning to combat the "negative transfer" phenomenon.

Experiments on the large HPO-B benchmark, as well as ablations (on synthetic few-shot regression problems, and in a non-transfer setting), are well-designed and support the paper's conclusions. The research question is clearly stated.

Weaknesses
==========
One would expect that the transfer/few-shot learning setting may be most useful with a small number of trials, however we see that the improvements (wrt. baselines) seem to only appear after about 10-20 trials, and are sustained afterwards. This may be investigated, or maybe addressed as a limitation in the context of the stated goal of limiting the number of HP configuration trials.

Minor points
==========
- Some confusion in notations in section 3.2
  * In Eq (1), is $K_x$ supposed to be $K_*$?
  * A mean function $m$ is introduced, but not used, and Eq (1) shows the mean as $0$.
  * In Eq (2), $y$ vs. ${\mathbf y}$
- It would be nice to keep the colors consistent between Figures 2 and 3 for the common curves

**Summary Of The Paper:**

The paper tackles the issue of speeding up black-box hyper-parameter optimization (HPO) by leveraging results obtained by trials on different datasets. It casts the problem into a few-shot regression one, where each task is a dataset and the goal is to predict the performance of a model trained on this dataset from the values of hyper-parameters. Meta-learning (REPTILE) is then used to initialize the (meta-)parameters of a deep kernel Gaussian process. Experiments on the HPO-B-v3 benchmark show this procedure equals or beats 10 baselines when using 20 to 100 trials.

**Summary Of The Review:**

Overall good, well-written, self-contained article that presents a novel combination of existing techniques, and addresses the issue of negative transfer when using transfer learning for Bayesian HPO. The experiments conducted are well designed and support the conclusion.

---

> ### Comment · Area_Chair_5eyF · 2021-11-10
> **elaborate on the potential impact of this submission?**
>
> Reviewer,
> Could you elaborate a bit on the potential impact of this submission? Whose life would change because of this paper existing/not existing? How would it change?

---

> > ### Comment · Reviewer_kJb9 · 2021-11-23
> > **Impact**
> >
> > Black-box hyper-parameter optimization is an important topic, as exploration of HPs still relies widely on personal experience and heuristics from practitioners, leading to 2 major issues:
> > - computation (and energy, CO2) wasted on inefficient exploration, and
> > - sub-optimal results, due to regions of interests in the HP space being left un-explored (in general, or within a resource or time budget).
> >
> > Another reviewer also mensions that:
> > > The topic is important. Hyperparameter is expensive and transfer learning is a promising solution to reduce the cost of HPO.
> >
> > I think this paper has the potential of making it easier:
> > - for ML practitioners to get good performance when using existing algorithms on data not dissimilar to the datasets used for HPO-B-v3
> > - for ML researchers to properly optimize baselines and explore novel algorithms on existing benchmarks (i.e., HPO-B-v3 or other HPO meta-datasets they would have access to).
> >
> > Even if there is not a net environmental impact (as people could decide to keep using the same amount of resources), it seems wasteful to not leverage data from past HPO optimizations, and this paper seems to be one of the best current ways of doing so.

---

> ### Author Response · Authors · 2021-11-17
> **Rebuttal**
>
> We would like to thank the reviewer for his interest in our paper and for his recommendation.
>
> We have fixed the minor confusions that you pointed out.
>
> We are happy to address any further remarks.

---

### Official Review · Reviewer_bzFz · 2021-11-03

**Correctness:** 3
**Technical Novelty And Significance:** 3
**Empirical Novelty And Significance:** 3
**Recommendation:** 6
**Confidence:** 3

**Main Review:**

Strengths:
- Adequately builds upon previous work such that the proposed change (the meta-extractor within the kernel of a deep GP) is easy to assess and compare (as opposed to larger set of changes that are not adequate ablated).
- Large-scale benchmark (935 tasks, 16 search spaces) compared to 10 baselines.
- Ablations with and within a randomly-initialized meta-extractor.
- Results included error bars (Table 1).

Weaknesses:
- Experimental setup could be made clearer as to which tasks are used for training and eval in the transfer setting (i.e., which tasks are used to originally train the meta-extractor, and on which tasks is it transferred).

Minor:
- p. 3: In the fourth line from the bottom, the reference "2020" needs to be fixed.

**Summary Of The Paper:**

The authors focus on the important problem of more efficient hyperparamter optimization through (meta) transfer learning. More specifically, the author propose a new method (Deep Kernel Gaussian Process surrogate with Landmark Meta-features (DKLM)) that embeds a meta-feature extractor model within the kernel function of a deep GP. This meta-feature extractor is able to learn to extract characteristics of a dataset such that the similarity of a dataset to other ones can be (implicitly) captured and used by the rest of the model. To demonstrate the effectiveness of their proposed model, they evaluate their model on the large-scale HPO-B-v3 benchmark against 10 other baselines plus a variant of their method with a randomly-initialized meta extractor.

**Summary Of The Review:**

Overall, the authors propose a new method that is well-scoped within existing literature, and provide positive empirical results on a large-scale benchmark against an adequate set of baseline tasks. It is a reasonable paper for acceptance.

---

> ### Author Response · Authors · 2021-11-17
> **Rebuttal**
>
> Thank you for your feedback.
>
> - We follow the same experimental protocol suggested by “HPO-B”. The benchmark paper introduces a predefined meta-train/meta-validation/meta-test split to facilitate transfer learning. The benchmark paper also evaluates a library of baselines, which allows us to compare directly to, considering that we adhere to the same protocol. We would like to note the meta-feature extractor is trained end-to-end and not separately.
>
> - We have added to Appendix B the detailed experimental protocol of HPO-B.
>
> - The mentioned reference is also fixed.
>
> We hope that this improves the chances of receiving a higher score as pointed out in your summary of the review.
>
> References:
>
> HPO-B: Sebastian Pineda-Arango, Hadi S. Jomaa, Martin Wistuba, and Josif Grabocka. HPO-B: A largescale reproducible benchmark for black-box HPO based on openml. Accepted at NeurIPS Datasets and Benchmark Track, 2021.

---

### Official Review · Reviewer_ZdAZ · 2021-11-04

**Correctness:** 3
**Technical Novelty And Significance:** 3
**Empirical Novelty And Significance:** 2
**Recommendation:** 5
**Confidence:** 3

**Details Of Ethics Concerns:**

I do not see any issues for ethics.

**Main Review:**

Strengths:
- The topic is important. Hyperparameter is expensive and transfer learning is a promising solution to reduce the cost of HPO.
- The solution is relatively complex, involving deep Kernels using neural networks and trained with meta-learning, and yet it is presented clearly and is easy to understand.
- The benchmark is large, including 10 different relevant algorithms on 935 tasks with 16 different search spaces.
- The experiments cover different aspects of the contributed algorithm; T-SNE projections of the meta-features and the impact of the sample size.

Weaknesses:
- A lot of emphasize in introduction is on the fact that DKLM would address negative transfer. I do not see this discussed explicitly in any of the experiments. If it is important, it should be supported by the experiments.
- The proposed solution is most certainly more computationally expensive than random search, or the fast Bayesian Optimization algorithm ABLR. The experiments should also show the improvement in terms of running time so that we can see how it performs compared to these faster algorithm.
- The standard deviation on Table 1 is way too large to be able to conclude anything. I could run again the same experiments with different seeds and observe an upward trend instead.
- The functions f and g introduces many hyperparameters to DKLM. The paper does not discuss how to chose these hyperparameters and whether tuning them is important for the performance of the algorithm.

Minor comments:
What is a meta-dataset? It is not defined in section 2.

I don’t understand this sentence:
> Consequently, the solution of the joint model resides on a local minimum so that given limited information about the new (unseen) target task.

Typos:
new a target task -> a new target task
fast adaption -> fast adaptation
where we notice -> while we notice

**Summary Of The Paper:**

This paper proposes a new Kernel for Bayesian Optimization to incorporate 'Landmark meta-features' of groups of tasks, so that the Kernel can be used to improve the efficience of the Bayesian Optimization on a new related task. The Kernel is based on 2 nested neural networks that are jointly trained via first-order meta-learning method REPTILE. The authors show that the method is overall better performing on the benchmark HPO-B-v3.

**Summary Of The Review:**

The proposed algorithm is an interesting way of learning end-to-end meta-features across tasks to improve the efficiency of Bayesian Optimization on new similar tasks. The benchmarking experiments are extensive and convincing for the efficiency of the algorithm, but does not support clearly the claims about negative transfer. The latter point should be discussed more clearly. The additional hyperparameters of the algorithms (network size and REPTILE hyperparameters) should also analyzed empirically to see how importantly they affect the performance of the algorithm.

---

> ### Author Response · Authors · 2021-11-17
> **Rebuttal**
>
> We thank the reviewer for his feedback. We address your concerns below.
>
> - We have added a description of negative transfer phenomena and how it might occur in the context of HPO in the introduction. We designed a new negative transfer experiment to test the performance of our method vs. baselines (Section 6.5) whereby the methods are meta-trained on uncorrelated tasks.
>
> - We agree that the implementation of DKLM is more computationally expensive than random search, since the latter is a simple model free approach. However, more recent transfer learning algorithms, such as ABLR, CTS, FSBO and ours, are composed of two stages: i) meta-learning the initialization of the surrogate on a collection of source tasks, ii) adapting the surrogate to the target task. For black-box optimization, it is common to evaluate the performance of the algorithms with respect to the number of trials and not w.r.t wall clock time. However, fitting our surrogate is fast. It takes on average (mean of all datasets in all search spaces) 9.7 milliseconds/epoch to fine-tune DKLM to new observations, and only 93.34 seconds on average to run our method for 100 trials across the 16 spaces.
>
> - We have pushed Section 6.5 (old) to the Appendix C and replaced it with a new experiment designed to address negative transfer. We have updated the table, however, to better illustrate the trend. We fit a linear model to the 10000 data points that have been sampled and report the slope. The negative slope indicates that the joint response surface is improved with more data points.
>
> -  DKLM is decomposed of two modules, $\phi_2$ with two multi-layer perceptrons f and g, and $\phi_1$ as another multi-layer perceptron which introduces the following hyper-hyperparameters: the number of layers, hidden units, activation function, dropout rate and learning rate.  We tune these hyper-hyperparameters through a random search on the total negative likelihood loss of the joint set of validation tasks.
>
> - We adopted a commonly-accepted terminology like prior work from HPO-B where a meta-dataset is a collection of hyperparameters and their responses that has been evaluated offline and collected in a tabular form to expedite transfer learning During training/inference, the response is simply queried for the suggested hyperparameters.
>
> - We have reformulated this sentence for better understanding.
>
> Please feel free to let us know if any questions or concerns still persist, that would prevent you from raising your score toward acceptance.
>
>
> References:
>
> HPO-B: Sebastian Pineda-Arango, Hadi S. Jomaa, Martin Wistuba, and Josif Grabocka. HPO-B: A largescale reproducible benchmark for black-box HPO based on openml. Accepted at NeurIPS Datasets and Benchmark Track, 2021.

---

> > ### Comment · Reviewer_ZdAZ · 2021-12-03
> > **Re: rebuttal**
> >
> > I thank the author for their detailed response.
> >
> > - I believe the experiments of section 6.5 is a reasonable way of measuring the impact of negative transfer. I disagree however that the results support the claim of the paper. On the mean normalized regret FSBO and DKLM using the decorrelated source tasks are about equivalent.
> > - It is true that hyperparameter optimization algorithms are rarely evaluated with respect to wall-clock time performance. One of the reason being that trial execution is typically much longer that the execution of the algorithm to suggest a new trial. It may not be true however when leveraging large amount of data for transfer learning in HPO. In particular, the paper presenting ABLR does evaluate wall-clock time performance because ABLR is a fast alternatives to other GP approaches for BO. Among other things, they show that BOHAMIAN performs better than ABLR for a given number of trials on a target task, but it is significantly slower on large set of trials for source tasks making it hardly usable for transfer learning in practice. This is why I am concerned with wall-clock time efficiency here.
> > - The slope measurement from one experiment to another is noisy. If we compute the confidence interval for the 16 reported experiments we get [-0.23, 0.1]. That is not significant.
> > ```
> > a = numpy.array([-0.075, -0.15, -0.077, -1.473, 0.215, -0.039, 0.414, -0.003, -0.015, 0.021, 0.14, -0.118, -0.032, 0.019, -0.08, 0.229])
> > confidence_interval = a.mean() + numpy.array([-1, 1]) * scipy.stats.t.ppf(0.95, df=len(a)) * a.std() / numpy.sqrt(len(a))
> > ```
> > - These experiments to select the hyperparameters of DKLM should be described. Sensitivity to hyperparameters of the HPO algorithm is important.

---

> > > ### Author Response · Authors · 2021-12-06
> > > **Rebuttal**
> > >
> > > Thank you again for engaging the in discussion:
> > >
> > > - In addition to the mean normalized regret, we present the critical difference diagram which explicitly shows that DKLM (NT) is statistically significant compared to FSBO (NT)
> > > - Thank you for point out the evaluation of ABLR w.r.t. to wall-clock time. We have reported the time to fine-tune DKLM in the previous rebuttal, and would like to note that even if ABLR might be faster compared to other BO approaches, the results are much worse than state-of-the-art. Since we are mainly concerned with optimizing the black-box objective, as do most of the baselines, we do not emphasize time.
> > > - We have moved 6.5 (old) to the appendix to avoid any confusion. The experiment was intended to highlight the improved performance of the surrogate with more observations after meta-initialization.

---

### Author Response · Authors · 2021-11-17
**Summary of Changes**

# Handling the major criticism on Negative Transfer

In the first round of reviews, almost all the reviewers raised criticism that a major claim of the paper in terms of handling negative transfer was not thoroughly supported empirically.
Although we demonstrated that our method DKLM outperforms a large set of state-of-the-art transfer HPO baselines, we agree that our experimental protocol was not specifically designed to test negative transfer per se.

To remedy this deficiency, we designed a negative transfer experiment as follows:

- For each search space in the HPO-B benchmark we reduce the set of meta-training datasets to only 10% of the most uncorrelated datasets to the meta-test datasets. As a result, competing methods would learn to transfer from training datasets that are uncorrelated with the testing datasets.

- Here, the correlation between a pair of datasets was measured as the rank correlation (Kendall tau) of the vector of validation accuracies achieved by evaluating the same set of random hyperparameter configurations on both datasets.

For more details, we added a new section 6.5 as "Resilience to Negative Transfer". The outcome of the experiment in Section 6.5 validates that our method DKLM with meta-features provides resilience to the negative transfer, by having a larger gap in performance to the variant without meta-features (FSBO) in the negative transfer experimental setup.

# Experimental Details
We also uploaded the source code, added two new algorithms that describe DKLM in pseudo-code, and added the experimental protocol in Appendix B.

We thank all the reviewers for their feedback, and are available to answer any additional concerns.

---

### Decision · Program_Chairs · 2022-01-20

**Decision:**

Reject

**Comment:**

The submission describes a method for tuning machine learning pipeline hyperparameters using transfer learning from related tuning tasks. In particular, the method uses learned meta features to construct a covariance function for a GP.

This was an extremely difficult case and could have gone either way. It was the closest case for any paper I serve as the AC for. Two of the reviewers recommended rejecting the paper and three recommended accepting, although during discussion one of the reviewers recommending accepting the paper seemed to actually be more on the reject side.

Ultimately, I have decided to recommend rejecting this submission. However, if either the clarity (especially concerning the neural network setup) or the experiments were somewhat improved I would have recommended accepting it. I view clarity as an extremely important factor when weighing whether a submission should be accepted. I concur with the reviewers on the following weaknesses of the experiments: (1) the lack of an ablation test when considering ad-hoc meta features and (2) the experimental evaluation is based on mostly aggregated metrics.

I know this recommendation must be disappointing, but I encourage the authors to polish the work a bit more and resubmit it somewhere.